# Two-Stage Ligand-Protein Complex Sequence Design with L-Caliby

**Jinho Kim**
Department of Physics
Stanford University
zhkim216@stanford.edu

**Richard W. Shuai**
Biophysics Program
Stanford University
rshuai@stanford.edu

**Po-Ssu Huang**[†]
Department of Bioengineering
Stanford University
possu@stanford.edu

## Abstract

Designing proteins that bind small-molecule ligands requires sequence design methods that account for ligand interactions while maintaining robust backbone foldability. A recent work, Caliby, has demonstrated that Potts model-based sequence design achieves state-of-the-art AlphaFold2 self-consistency for protein structural design, without features to model protein-ligand interactions. Here, we introduce L-Caliby, a Potts model for ligand-conditioned protein sequence design that augments the Potts-based architecture with LigandMPNN's ligand encoder and protein-ligand interaction module. L-Caliby employs a two-stage pocket-then-scaffold (Pocket→Scaffold) design paradigm using two independently trained design models: a pocket model that optimizes protein-ligand interactions, and a scaffold model that ensures robust foldability of the remaining sequence. On 61 held-out native ligand-protein complexes evaluated by AlphaFold3 in single-sequence mode, L-Caliby (Pocket→Scaffold) outperforms LigandMPNN on ligand metrics by more than fourfold. On 1,220 de novo protein-ligand complexes generated by RFdiffusion3 with unseen ligands, L-Caliby similarly outperforms LigandMPNN by 28% on PoseBusters-valid ligand success rate. These results reveal that a two-stage Potts model with ligand conditioning can satisfy the different requirements needed for pocket and scaffold redesign tasks and simultaneously optimize both.

## 1 Introduction

The design of proteins that bind small-molecule ligands is central to biosensor design and enzyme design (Butcher et al., 2025), requiring a method to design sequences that accounts for the interactions with a variety of ligands while maintaining the overall structural integrity of the protein. While there are a few co-design models (Didi et al., 2026; Wang et al., 2026; Butcher et al., 2025) that can simultaneously generate protein structure, sequence, and sidechains given a ligand context, structure-conditioned sequence design models, such as LigandMPNN (LMPNN) (Dauparas et al., 2025), a widely-used method for ligand-conditioned sequence design, consistently outperform these co-design models when the designed sequences are evaluated by structure prediction models based on the accuracy of protein and ligand structural recovery, defined as self-consistency (Wang et al., 2026). However, LMPNN-produced designs frequently fail at self-consistency trials on both native and *de novo* backbones when evaluated by AF3, with median self-consistency RMSD (scRMSD) values exceeding 14 Å and 3 Å, respectively.

Recent works have shown that Potts model-based sequence design (Shuai et al., 2025; Birnbaum & Keating, 2026) outperforms autoregressive approaches on self-consistency, suggesting a better inductive bias for structure-sequence relationships. However, these methods currently lack ligand

---

[†]Corresponding author: possu@stanford.edu

conditioning. This motivates our work to leverage the Potts model's inductive bias for ligand-conditioned design, combining its generalizability with explicit ligand awareness.

Our key contributions are as follows:

- We conduct extensive evaluation of how Potts models behave under varying backbone representations, noise levels, and training steps, revealing that foldability and ligand binding quality respond in opposite directions to these hyperparameters. Based on this analysis, we propose a two-stage pocket-then-scaffold paradigm that resolves this trade-off.
- We introduce ligand-conditioned Potts models that integrate LigandMPNN's ligand encoder and protein-ligand interaction module into the Potts-based architecture.
- We curate evaluation benchmarks designed for rigorous assessment of ligand-binding protein sequence design, addressing protein chain cluster overlap and ensuring ligand novelty and diversity.
- Combining these contributions, we present L-Caliby, which achieves state-of-the-art performance on both native and de novo ligand-binding protein benchmarks as evaluated by AF3.

## 2 BACKGROUND AND RELATED WORK

**Potts models and structure prediction for evaluation.**  The mapping from backbone structure to amino acid sequence lies at the core of protein sequence design. Potts models parameterize the joint distribution over sequences as a pairwise Markov random field, capturing both single-site preferences and pairwise residue-residue interactions. This formulation provides a natural inductive bias for modeling structure-sequence relationships, as the identity of each residue depends on the identities of spatially neighboring residues through pairwise couplings. Structure prediction models such as AF2 (Jumper et al., 2021) and AF3 (Abramson et al., 2024) have enabled *in silico* evaluation of designed sequences through self-consistency: a designed sequence is provided to a structure predictor, and the predicted structure is compared against the input backbone. This self-consistency pipeline provides a stringent test of whether a designed sequence can actually fold into the intended structure. AF3 further enables evaluation of ligand placement by jointly predicting protein structure and ligand pose, making it possible to assess whether a designed sequence encodes both the target fold and the intended binding pocket.

**Related work.**  Caliby (Shuai et al., 2025) demonstrated that Potts model-based sequence design achieves state-of-the-art AF2 self-consistency for protein-only backbones, outperforming the autoregressive ProteinMPNN (Dauparas et al., 2022). LigandMPNN (Dauparas et al., 2025) extends ProteinMPNN with atomic-level ligand context, enabling ligand-conditioned sequence design through an autoregressive architecture. While LigandMPNN achieves high native sequence recovery, it frequently yields poor self-consistency when evaluated by AF3. PottsMPNN (Birnbaum & Keating, 2026), a concurrent method, also builds on Potts model-based sequence design and provides evidence that explicit pairwise residue interactions are critical for capturing structure-sequence relationships. LASErMPNN (Fry et al., 2025) trains a ligand-aware GNN with a pre-trained ligand encoder, co-generating sequences and sidechain conformations including hydrogens, supervised with an MSA-based loss, and pairs it with an iterative selection-expansion loop for zero-shot drug-binding protein design. ADFLIP (Yi et al., 2025) applies discrete flow matching to all-atom inverse folding, conditioning on ligands, nucleotides, and metal ions while progressively incorporating predicted sidechains during sampling. L-Caliby differs from these approaches by combining Potts model-based design with ligand conditioning and introducing a two-stage paradigm that separately optimizes foldability and ligand binding quality.

## 3 MODEL ARCHITECTURE

L-Caliby builds on the Potts model-based sequence design model architecture (Ingraham et al., 2023; Shuai et al., 2025), augmented with the ligand encoder and protein-ligand interaction module from LMPNN (Dauparas et al., 2025). The model consists of three stages: (1) three parallel encoders that jointly produce residue-level embeddings with each residue attending to its $k$ nearest protein

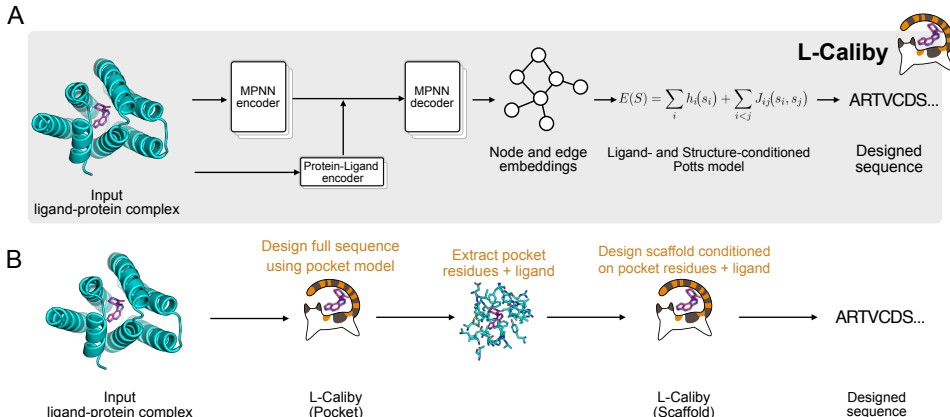

Figure 1: **Overview of L-Caliby. (A)** L-Caliby architecture: an MPNN encoder and a protein-ligand encoder jointly feed into an MPNN decoder to produce node and edge embeddings, which parameterize a ligand- and structure-conditioned Potts model for sequence design. **(B)** Two-stage pocket-first design (P→S): the pocket model first designs the full sequence, then pocket residues and the ligand are extracted and passed to the scaffold model, which redesigns the remaining residues conditioned on the pocket residues and ligand.

neighbors and $m$ nearest ligand atoms: (a) a protein backbone encoder that takes pairwise distances between backbone atoms ($C\alpha$, $NC\alpha CO$, or $NC\alpha COC\beta$, depending on the configuration) as input, (b) a ligand encoder that operates on pairwise distances and element types among ligand atoms, and (c) a protein-ligand interaction encoder that uses pairwise distances and angle features between ligand atoms and $NC\alpha COC\beta$ backbone atoms along with element information, (2) an MPNN decoder that refines these embeddings, where we remove the causal mask used in standard MPNN, and (3) a Potts head that transforms the decoder outputs into single-site biases $h_i \in \mathbb{R}^{20}$ and pairwise couplings $J_{ij} \in \mathbb{R}^{20 \times 20}$ for all residue pairs within a spatial neighborhood, defining a pairwise Markov random field:

$$P(\mathbf{s} \mid \mathbf{X}, \mathbf{L}) \propto \exp\left(\sum_i h_i(s_i) + \sum_{i<j} J_{ij}(s_i, s_j)\right), \tag{1}$$

where $\mathbf{s}$ is the amino acid sequence, $\mathbf{X}$ is the backbone structure, and $\mathbf{L}$ represents the ligand coordinates and features. Sequences are sampled from this distribution using discrete Langevin Monte Carlo (DLMC) sampling (Ingraham et al., 2023).

## 4 EVALUATION METRICS

**Metrics.** We evaluate designed sequences along two primary axes, backbone foldability and ligand quality, and report sequence recovery as an additional monitoring metric. All metrics are computed from AF3 self-consistency evaluations: each designed sequence is provided to AF3 together with the ligand's chemical identity, without the 3D ligand coordinates, in single-sequence mode (no multiple sequence alignment, no backbone template, 10 recycles), removing MSA-derived evolutionary signal and reducing the influence of similar pockets in AF3's training data (Škrinjar et al., 2025).

*Backbone foldability* measures how well the designed sequence encodes the intended fold. **$C\alpha$ self-consistency RMSD** (scRMSD, ↓) measures the structural deviation between predicted and input backbones after $C\alpha$ superposition, where lower values indicate that the designed sequence successfully encodes the target fold. **$C\alpha$ pLDDT** (↑) reflects AF3's per-atom confidence in its backbone prediction, serving as a proxy for how confidently the sequence can be folded into the intended structure.

*Ligand quality* assesses whether the designed protein can accommodate the target ligand in the correct pocket and pose, comprising per-sample metrics, composite success rates, and benchmark-level coverage. **Pocket-aligned ligand RMSD** ($\downarrow$) measures the deviation of the predicted ligand pose from the reference pose after superposition on pocket residues, capturing how well the designed binding pocket preserves the intended ligand geometry. Unless otherwise stated, ligand RMSD refers to pocket-aligned ligand RMSD throughout. **Ligand pLDDT** ($\uparrow$) reflects AF3's confidence in the predicted ligand placement, indicating whether the designed pocket residues can hold the ligand. **Ligand placement rate** ($\uparrow$) reports the percentage of complexes that simultaneously achieve a well-folded backbone and precise ligand placement. Specifically, a design is counted as a successful placement if: (1) the protein backbone is well-folded (scRMSD $\leq 2\,\text{Å}$ and C$\alpha$ pLDDT $\geq 80$ for chains with length $\leq 250$, or scRMSD $\leq 3\,\text{Å}$ and C$\alpha$ pLDDT $\geq 70$ for longer chains), and (2) the ligand is placed precisely (pocket-aligned ligand RMSD $\leq 2\,\text{Å}$). This joint criterion ensures that we evaluate ligand placement only for designs where the backbone is confidently folded, disentangling ligand quality from foldability. **Ligand success rate** ($\uparrow$) additionally requires ligand pLDDT $\geq 80$, providing a joint test of backbone foldability, precise ligand positioning, and high ligand prediction confidence. Preliminary experimental evidence from Fry et al. (2025) suggests that AF3 ligand pLDDT $\geq 80$ separates successful binders from failures and correlates with binding affinity, though this was observed for a few experimental examples. We adopt this threshold as our ligand pLDDT cutoff accordingly. We adopt ligand placement rate and ligand success rate as the primary ligand metrics throughout this work, as median ligand RMSD, median ligand pLDDT, and conditional ligand pLDDT (e.g., median ligand pLDDT given placement or given success) can be misleading when only a small number of samples passed each of the filters (Appendix H). We also report results at additional ligand RMSD and pLDDT cutoff combinations in Appendix G. **CCD coverage** ($\uparrow$) reports the fraction of the 61 unique CCD codes for which at least one de novo design achieves ligand success across all backbone samples (five backbones $\times$ four length bins per CCD code). Because de novo backbones are generated without guarantees on pocket geometry, a single successful design per CCD code is sufficient to demonstrate that the model can accommodate that ligand. This metric complements per-sample success rates by quantifying the breadth of ligand types a model can design for.

*Sequence recovery* quantifies the agreement of the designed sequence with the native sequence. Overall sequence recovery reports the fraction of residues matching the native sequence across the full protein, while pocket sequence recovery restricts this calculation to residues with at least one heavy atom within $4\,\text{Å}$, $5\,\text{Å}$, or $6\,\text{Å}$ of the ligand. Sequence recovery is known to be anti-correlated with self-consistency for overall protein sequences (Shuai et al., 2025), and thus is a poor proxy for foldability. Whether pocket sequence recovery correlates with ligand placement quality remains an open question. We report pocket sequence recovery as an additional monitoring metric alongside the primary backbone foldability and ligand quality metrics.

**Benchmarks.** We evaluate on two complementary benchmarks:

**Nativeval** comprises 61 native protein-ligand complexes curated from the LigandMPNN validation set and structures deposited after the training cutoff, filtered to ensure protein chain cluster diversity, ligand novelty (Tanimoto similarity $< 0.8$ to training ligands), and ligand chemical diversity. For each complex, we sample 4 sequences per method and report the best-of-4 result.

**Denovoval** comprises 1,220 de novo protein-ligand complexes generated by RFD3 (Butcher et al., 2025) using CCD codes selected from nativeval, with 5 backbone samples at each of 4 lengths (150, 250, 350, 450 residues) per ligand. For each complex, we sample 2 sequences per method and report the best-of-2 result.

We additionally define designable subsets of each benchmark (nativeval-designable, denovoval-designable) for hyperparameter tuning of the pocket model. Additional details on benchmark set curations are provided in Appendix A and Appendix B, and details on AF3 evaluation setup and reporting conventions are provided in Appendix D.

# 5 RESULTS

## 5.1 TWO-STAGE DESIGN TO RESOLVE THE TRADE-OFF BETWEEN LOCAL ATOMIC ACCURACY AND GLOBAL FOLDABILITY

Through systematic ablations on backbone representation, training noise, and training duration using 61 held-out native and 1,220 RFdiffusion3 de novo ligand-protein complexes, we found that local atomic accuracy at the binding site and global foldability of the scaffold respond in opposite directions to these hyperparameters. Pocket residues, which must form precise geometric and chemical complementarity with the ligand, benefit from richer atomic representations and lower coordinate noise that preserve the detailed binding site geometry. Scaffold residues, whose primary requirement is to encode a globally foldable topology, instead benefit from coarser representations or higher noise that regularize against local overfitting and promote capture of fold-level features. This trade-off motivates a two-stage design approach.

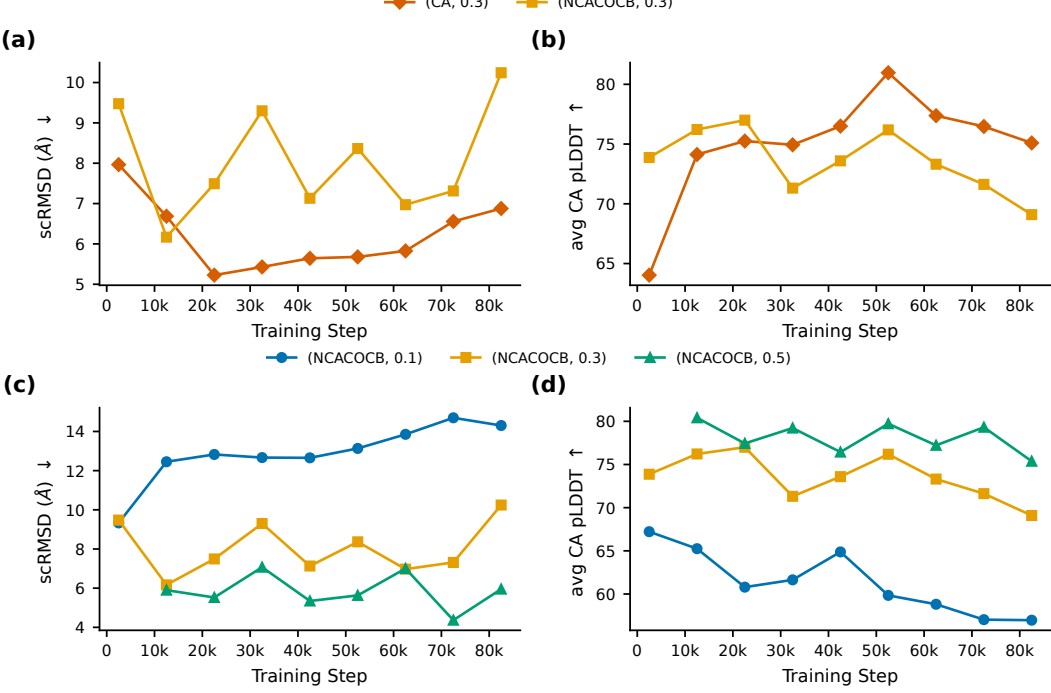

Figure 2: **Foldability of ligand-conditioned Potts models on nativeval.** **(a)** scRMSD and **(b)** C$\alpha$ pLDDT across training steps for three backbone representations (C$\alpha$, NC$\alpha$CO, NC$\alpha$COC$\beta$) at $\sigma = 0.3$ Å. **(c)** scRMSD and **(d)** C$\alpha$ pLDDT across training steps for NC$\alpha$COC$\beta$ at varying noise levels ($\sigma \in \{0.1, 0.2, 0.3, 0.4, 0.5\}$ Å).

**Foldability.** We first assess how backbone representation and training noise affect foldability, measured by scRMSD and C$\alpha$ pLDDT. Figure 2 compares ligand-conditioned Potts models on nativeval under these varying conditions.

On nativeval (Figure 2a,b), the backbone representation comparison at $\sigma = 0.3$ Å reveals that the C$\alpha$-only model achieves the best C$\alpha$ pLDDT across training, while the richer NC$\alpha$COC$\beta$ representation overfits, with C$\alpha$ pLDDT degrading as training progresses. For the noise level comparison on NC$\alpha$COC$\beta$ (Figure 2c,d), higher noise ($\sigma = 0.5$ Å) yields the best foldability, maintaining stable performance throughout training, while lower noise ($\sigma = 0.1$ Å) leads to severe overfitting with scRMSD increasing and C$\alpha$ pLDDT dropping sharply.

On denovoval (Figure S2), the pattern reverses: the C$\alpha$-only model achieves the worst foldability, while NC$\alpha$CO ($\sigma = 0.3$) and NC$\alpha$COC$\beta$ ($\sigma = 0.3$) converge to substantially better scRMSD and

C$\alpha$ pLDDT. This reversal motivates the selection of NC$\alpha$COC$\beta$ ($\sigma = 0.5$) as the scaffold model (Appendix E), as it achieves top foldability on denovoval while maintaining adequate performance on nativeval.

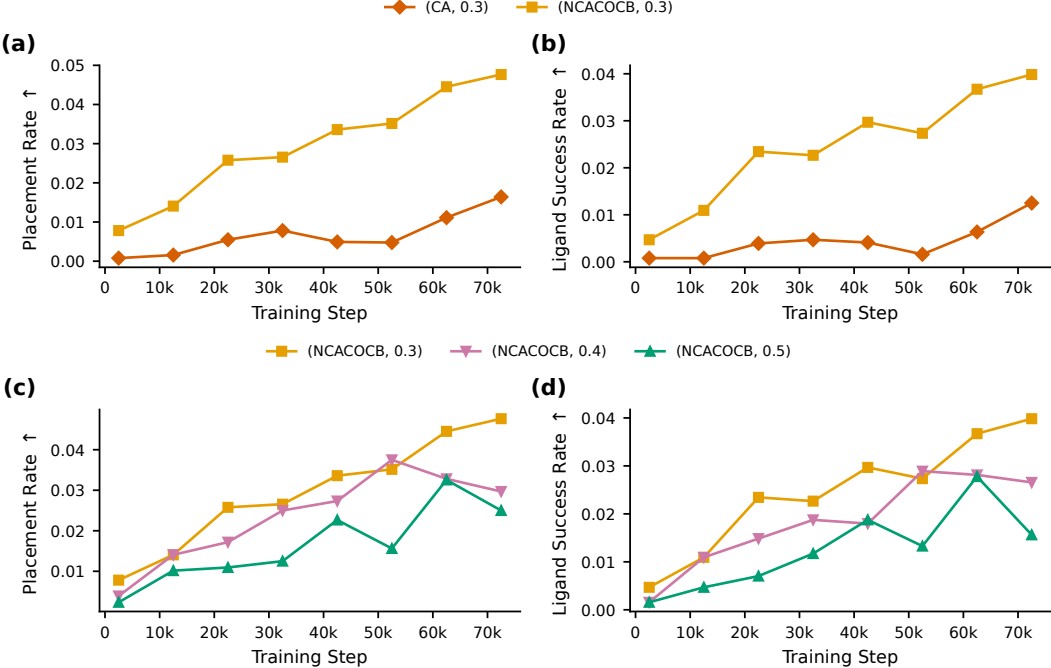

Figure 3: **Pocket sequence quality of ligand-conditioned Potts models on denovoval.** **(a)** Ligand placement rate and **(b)** ligand success rate across training steps for NC$\alpha$COC$\beta$ at $\sigma \in \{0.3, 0.4, 0.5\}$ Å. **(c)** Ligand placement rate and **(d)** ligand success rate across training steps for three backbone representations (C$\alpha$, NC$\alpha$CO, NC$\alpha$COC$\beta$) at $\sigma = 0.3$ Å.

**Pocket sequence quality.** The foldability trends are opposed by the pocket sequence quality trends. Figure 3 evaluates pocket sequence quality on denovoval using ligand placement rate and ligand success rate. For the noise level comparison (Figure 3a,b), NC$\alpha$COC$\beta$ models at $\sigma \in \{0.3, 0.4, 0.5\}$ Å are compared. Models with $\sigma \leq 0.2$ Å were excluded in the noise level comparison, as they significantly overfit on nativeval foldability. Among the evaluated noise levels, lower noise consistently improves pocket sequence quality, with $\sigma = 0.3$ Å outperforming $\sigma = 0.5$ Å on both metrics. For the backbone representation comparison (Figure 3c,d), richer representations consistently improve pocket quality, with NC$\alpha$COC$\beta$ outperforming NC$\alpha$CO and C$\alpha$. Unlike foldability, pocket sequence quality improves continuously with training steps across all configurations, showing no sign of overfitting.

**Two-stage paradigm.** These opposing trends motivate a two-stage design paradigm. In the first stage, we design pocket residues using a model optimized for pocket quality: an NC$\alpha$COC$\beta$ representation with the lowest noise ($\sigma = 0.1$ Å), which provides detailed geometric information about the binding site. In the second stage, we fix the pocket sequence and design the remaining scaffold residues using a separately trained scaffold model (NC$\alpha$COC$\beta$, $\sigma = 0.5$ Å), selected based on the scaffold analysis in Figure S2. This pocket-then-scaffold approach allows L-Caliby to achieve high quality in both regions simultaneously, overcoming the trade-off that limits any single-model configuration. We define pocket residues as those with any heavy protein atom within a distance cutoff $d$ of any heavy ligand atom: $d = 6$ Å for nativeval and $d = 8$ Å (using pseudo-C$\beta$ coordinates) for denovoval, as sidechains generated by RFD3 could be unreliable. Details of the pocket distance sweep are provided in Appendix F.

Table 1: **AF3 evaluation on native ligand-protein complexes (nativeval, $n = 61$).** Pocket/Scaffold: each model applied independently to the full sequence. Nat. Pkt: native pocket + scaffold model. S→P / P→S: two-stage designs. All values are medians except rates. ↓: lower is better, ↑: higher is better. Best values are **bolded**.

| Method | scRMSD (Å, ↓) | C$\alpha$ pLDDT (↑) | Place Rate (↑) | Lig Succ Rate (↑) | Pkt Seq Rec (↑) | | |
| --- | --- | --- | --- | --- | --- | --- | --- |
| | | | | | 4 Å | 5 Å | 6 Å |
| Native | 18.49 | 40.0 | 1.6% | 1.6% | 100% | 100% | 100% |
| ProteinMPNN | 13.49 | 56.9 | 3.3% | 3.3% | 44.4% | 45.5% | 46.5% |
| LigandMPNN | 14.97 | 57.2 | 4.9% | 3.3% | 61.1% | 59.3% | 59.1% |
| L-Caliby (Scaffold) | 5.04 | 78.0 | 6.6% | 6.6% | 43.3% | 41.4% | 41.5% |
| L-Caliby (Pocket) | 13.02 | 59.2 | 9.8% | 8.2% | 60.6% | 61.1% | 58.3% |
| L-Caliby (Native Pocket) | 6.58 | 74.0 | 11.5% | 9.8% | 100% | 100% | 100% |
| L-Caliby (S→P) | 5.74 | 78.2 | 8.2% | 8.2% | 60.0% | 57.1% | 56.4% |
| L-Caliby (P→S) | **4.51** | **80.5** | **16.4%** | **14.8%** | 60.0% | 61.5% | 57.9% |

## 5.2 POCKET-FIRST DESIGN STRATEGY

We compared two orderings for the two-stage design: pocket-first (P→S, designing pocket residues first, then scaffold) and scaffold-first (S→P, designing scaffold residues first, then pocket). P→S designs the entire sequence first using the pocket model and then redesigns the scaffold sequence while keeping the pocket sequence fixed. This allows the pocket model to optimize protein-ligand interactions without constraints from a pre-determined scaffold sequence, while the scaffold model rescues the scaffold sequence that is likely unfoldable. S→P follows a similar procedure but uses the scaffold model first and then designs pocket sequences constrained on the designed scaffold sequence.

## 5.3 NATIVE PROTEIN-LIGAND COMPLEXES (NATIVEVAL)

We evaluated L-Caliby on 61 native protein-ligand complexes from nativeval, comparing against LigandMPNN and the native sequence baseline. We include three ablation baselines: the scaffold model applied to the full sequence, isolating its behavior without the pocket stage; the pocket model applied to the full sequence, isolating its behavior without the scaffold stage; and the native pocket baseline, which retains the native pocket sequence and designs only the scaffold residues using the scaffold model.

Figure 4 and Table 1 summarize the results. The native sequence, when evaluated by AF3, already shows poor self-consistency (scRMSD 18.49 Å, C$\alpha$ pLDDT 40.0). The pocket model and scaffold model exhibit complementary strengths: the pocket model achieves strong ligand success rate (8.2%) but poor foldability (scRMSD 13.02 Å, C$\alpha$ pLDDT 59.2), while the scaffold model achieves strong foldability (scRMSD 5.04 Å, C$\alpha$ pLDDT 78.0) but low ligand success rate (6.6%). The two-stage P→S model inherits the best of both: the lowest scRMSD (4.51 Å), the highest C$\alpha$ pLDDT (80.5), and the highest ligand success rate (14.8%). Compared to LigandMPNN, P→S improves ligand success rate by more than fourfold (14.8% vs. 3.3%) and scRMSD by over 10 Å (4.51 Å vs. 14.97 Å). Compared to S→P, P→S shows better ligand metrics. This is because P→S model can design pocket sequences without constraints from a pre-determined scaffold sequence.

Interestingly, the native pocket baseline shows ligand success rate (9.8%) lower than P→S (14.8%) despite retaining 100% pocket sequence recovery, an observation we discuss further in Section 6.

## 5.4 DE NOVO PROTEIN-LIGAND COMPLEXES (DENOVOVAL)

We evaluated L-Caliby on 1,220 de novo protein-ligand complexes from denovoval, comparing L-Caliby variants against ProteinMPNN, LigandMPNN, and the RFD3 backbone reference. As on nativeval, we include the pocket model and scaffold model applied independently to the full sequence as ablation baselines.

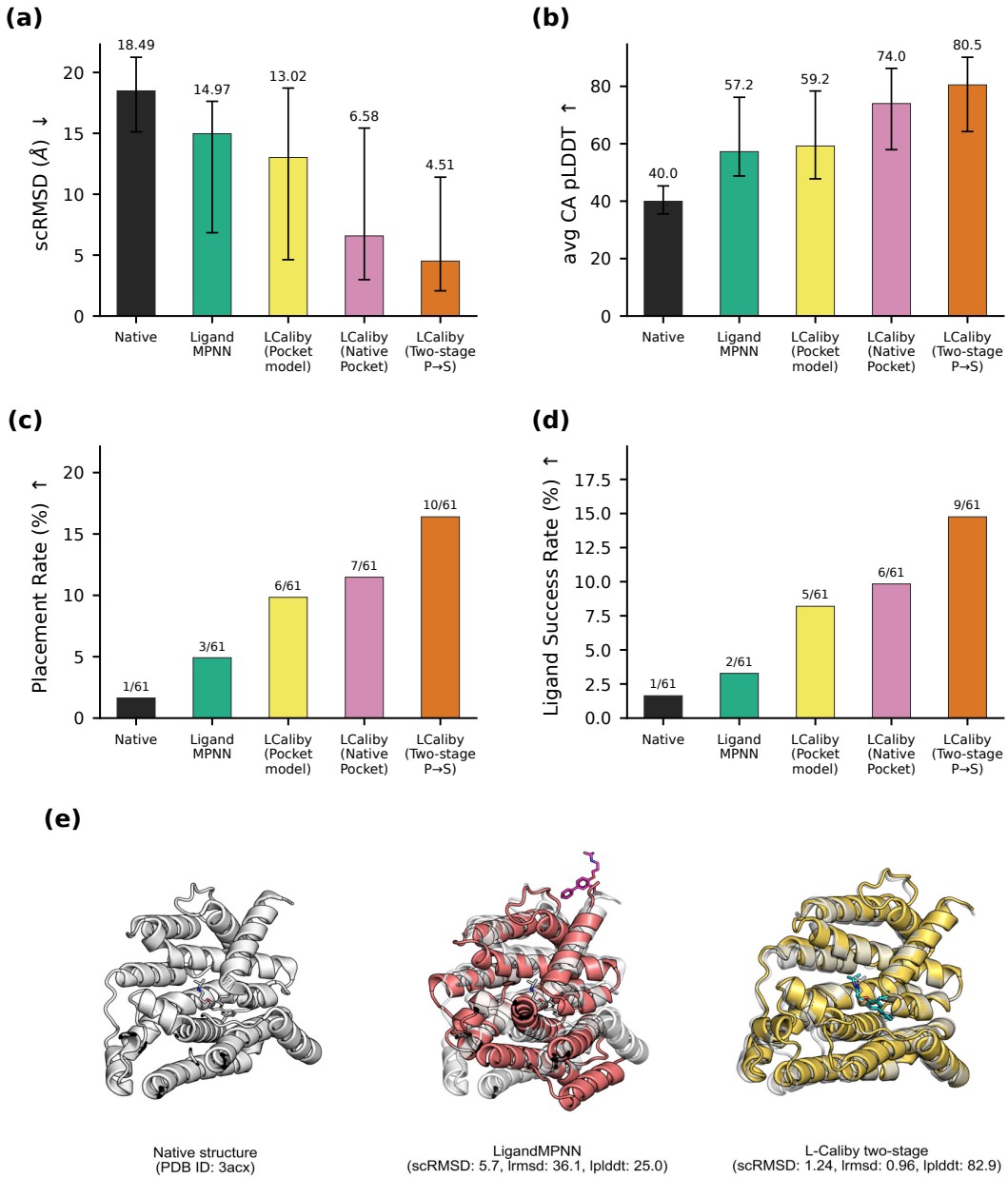

Figure 4: **AF3 evaluation on 61 native ligand-protein complexes (nativeval). (a)** Median scRMSD with interquartile range. **(b)** Median Cα pLDDT. **(c)** Ligand placement rate (pocket-aligned ligand RMSD ≤ 2 Å). Counts above bars show the number of successes out of 61. **(d)** Ligand success rate (placement rate with ligand pLDDT ≥ 80). Counts above bars show the number of successes out of 61. Median values are annotated above bars in (a) and (b). Arrows in the y-axis labels indicate the direction of improvement. **(e)** Example AF3 predictions for PDB 3acx. Left: native structure (gray). Middle: LigandMPNN design (red, scRMSD 5.7 Å, ligand RMSD 36.1 Å, ligand pLDDT 25.0). Right: L-Caliby P→S design (gold, scRMSD 1.24 Å, ligand RMSD 0.96 Å, ligand pLDDT 82.9). The ligand is shown in stick representation.

Figure 5 and Table 2 summarize the results. On foldability, the scaffold model achieves the best performance (scRMSD 1.41 Å, Cα pLDDT 91.0), followed by P→S (1.95 Å, 87.7) and the pocket model (2.23 Å, 86.5). All L-Caliby variants substantially improve over LigandMPNN (3.12 Å, 84.7), though the gap is narrower than on nativeval, as de novo backbones tend to be easier to fold (Shuai

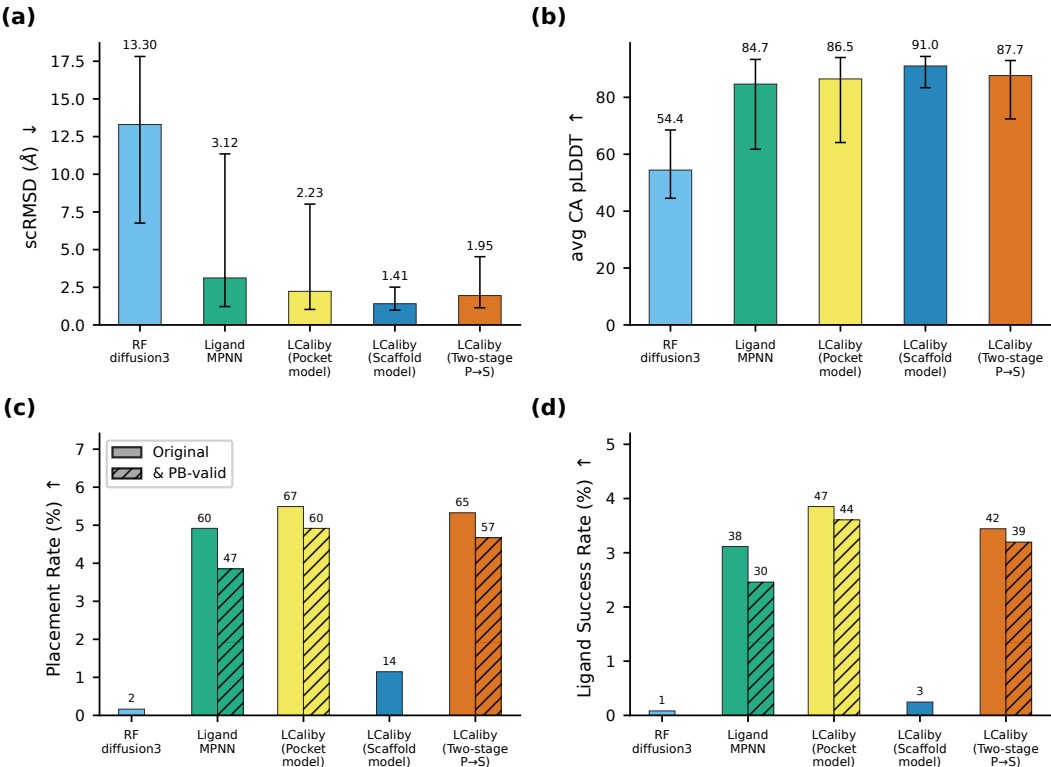

Figure 5: **AF3 and PoseBusters evaluation on 1,220 de novo ligand-protein complexes (denovo-val).** **(a)** Median scRMSD with interquartile range. **(b)** Median $C\alpha$ pLDDT. **(c)** Ligand placement rate (pocket-aligned ligand RMSD $\leq 2\,\text{Å}$). Hatched bars show PB-valid placement rate (placement pass and PoseBusters valid). Counts above bars show the number of successes out of 1,220. **(d)** Ligand success rate (placement rate with ligand pLDDT $\geq 80$). Hatched bars show PB-valid success rate (success pass and PoseBusters valid). Counts above bars show the number of successes out of 1,220. Median values are annotated above bars in (a) and (b). Arrows in the y-axis labels indicate the direction of improvement. PB-filtered rates are shown for L-Caliby (Pocket), L-Caliby (P→S), and LigandMPNN.

Table 2: **AF3 evaluation on de novo ligand-protein complexes (denovoval, $n = 1,220$).** Pocket/Scaffold: each model applied independently to the full sequence. S→P / P→S: two-stage designs. Lig Succ PB: PB-valid success rate. CCD Cov (PB): CCD coverage (after PB filtering). All values are medians except rates. Best values are **bolded**.

| Method | scRMSD (Å, ↓) | $C\alpha$ pLDDT (↑) | Place Rate (↑) | Lig Succ Rate (↑) | Lig Succ PB (↑) | CCD Cov (↑) | CCD Cov PB (↑) |
|---|---|---|---|---|---|---|---|
| RFD3 (backbone) | 13.30 | 54.4 | 0.2% | 0.1% | – | – | – |
| ProteinMPNN | 2.28 | 87.9 | 0.2% | 0.2% | – | – | – |
| LigandMPNN | 3.12 | 84.7 | 4.9% | 3.1% | 2.5% | 44.3% | 34.4% |
| L-Caliby (Scaffold) | **1.41** | **91.0** | 1.1% | 0.2% | – | – | – |
| L-Caliby (Pocket) | 2.23 | 86.5 | **5.5%** | **3.9%** | **3.6%** | **52.5%** | **49.2%** |
| L-Caliby (S→P) | 1.69 | 89.3 | 4.3% | 2.8% | – | – | – |
| L-Caliby (P→S) | 1.95 | 87.7 | 5.3% | 3.4% | 3.2% | 42.6% | 42.6% |

et al., 2025). The RFD3 backbone reference (scRMSD 13.3 Å) shows that it requires sequence optimization to achieve AF3-foldable structures.

On ligand metrics, both the pocket model and P→S outperform LigandMPNN, and, again, P→S outperforms S→P. Notably, the pocket model achieves the highest ligand placement rate (5.5%) and ligand success rate (3.9%), outperforming even P→S (5.3% placement, 3.4% success), which we discuss further in Section 6.

To validate that these improvements reflect genuine chemical quality rather than AF3 confidence biases, we applied PoseBusters (Buttenschoen et al., 2024) to AF3-predicted ligand poses (Figure 5c,d, lighter bars). Both the pocket model and P→S retain higher PB-valid success rates (3.6% and 3.2%) than LigandMPNN (2.5%). The gap between L-Caliby and LigandMPNN widens after PoseBusters filtering, suggesting that a larger fraction of L-Caliby's successful designs correspond to chemically valid ligand poses. In addition, both the pocket model and P→S achieve higher PB-filtered CCD coverage (49.2% and 42.6%) than LigandMPNN (34.4%), indicating that pocket model generalizes to a broader range of ligand types while maintaining chemically valid ligand poses. Full Pose-Busters results across cutoffs and datasets are reported in Appendix I, and breakdown analyses by ligand property and protein length are provided in Appendix J.

# 6 DISCUSSION AND FUTURE WORK

**Performance gap on de novo vs. native complexes.** L-Caliby's advantage over LigandMPNN is more pronounced on nativeval than on denovoval. On native backbones, which can contain irregular or challenging structural features, L-Caliby's advantage on both foldability and ligand metrics becomes more apparent. This also explains why the pocket model alone outperforms P→S on denovoval ligand metrics: at shorter lengths, the fold is simple enough that the pocket model can produce foldable sequences without a dedicated scaffold stage, while at longer lengths (350+) this advantage disappears (Appendix J).

**P→S outperforms the native pocket baseline.** L-Caliby (P→S) outperforms the native pocket baseline on ligand success rate (Table 1), despite the latter retaining 100% pocket sequence recovery. This gap may indicate the existence of a "designable pocket sequence distribution", or it may arise because the native pocket at the 6 Å cutoff encompasses residues beyond the immediate binding site, carrying evolutionary context incompatible with a de novo-designed scaffold sequence. Future work will investigate this effect in greater detail.

**Evaluation and benchmark limitations.** Our evaluation relies solely on AF3. While AF3 is widely adopted, it remains a learned predictor with its own biases. Complementing AF3 evaluation with physics-based methods would further strengthen the assessment. Additionally, our benchmarks do not include recently proposed ligand-aware sequence design methods such as LASErMPNN (Fry et al., 2025) and ADFLIP (Yi et al., 2025). Direct comparison with these methods under matched evaluation budgets, as well as physics-based validation, are directions for future work.

ACKNOWLEDGEMENTS

We thank Gina El Nesr for helpful feedback on the paper. The computation for this project was performed on the Sherlock cluster at Stanford University. R.W.S. acknowledges funding support from the NSF Graduate Research Fellowship (DGE-2146755). Additional support to P.-S.H. is from Merck Research Laboratories (MRL) Scientific Engagement and Emerging Discovery Science (SEEDS) Program, Stanford Medicine Catalyst, and NIH (R01GM147893). The views and conclusions contained in this document are those of the authors and should not be interpreted as representing the official policies, either expressed or implied, of the U.S. Government.

REPRODUCIBILITY STATEMENT

Code and trained model weights will be released upon acceptance.

USE OF LARGE LANGUAGE MODELS

Large language models were used to assist in drafting and editing this manuscript and in generating plotting scripts for figures. All outputs were reviewed and verified by the authors.

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

SUPPLEMENTARY MATERIAL

## A    DATA PROCESSING AND DATASET CURATION

**Data processing.**    All structural data were processed using the Atomworks framework.

**Training data.**    We downloaded all PDB structures released before 2025-12-15 and clustered protein chains at 40% sequence identity. Following LigandMPNN (Dauparas et al., 2025), the training set consists of structures released before 2022-12-16. All PDB entries whose chains belong to the same cluster as any chain in the LigandMPNN validation sets (small molecule, metal, and nucleotide) were excluded from training.

We applied the following metadata filters: release date $\leq$ 2022-12-16, resolution $< 9.0\,\text{Å}$, no severe clashes, method restricted to X-ray diffraction or electron microscopy, and fewer than 50 polymer entities. Protein monomer chains were required to have between 20 and 2,048 resolved residues. For protein-ligand interfaces, we additionally required resolution $< 3.5\,\text{Å}$.

**Nativeval curation.**    We curated nativeval from LigandMPNN small molecule validation set structures and structures deposited after 2022-12-16, ensuring that no protein chain cluster overlaps with the training set. This is because we found during model development that unbalanced protein chain cluster sampling weights lead to overfitting, and that the LigandMPNN validation set contains substantial protein chain cluster overlap with the training data.

Ligand quality was filtered following PLINDER (Durairaj et al., 2024) criteria: non-hydrogen atom count $> 5$, carbon atom count $> 2$, unbranched hydrocarbon linker length $\leq 12$, all atoms resolved, no alternative configurations, exclusion of PLINDER artifact ligands, absolute formal charge $\leq 2$, resolution $\leq 3.5\,\text{Å}$, RSCC $\geq 0.8$, RSR $\leq 0.3$, and average occupancy $\geq 0.8$ with no clashes. We then selected 61 complexes to maximize chemical diversity along three ligand property axes: number of heavy atoms, fraction of rotatable bonds (number of rotatable bonds divided by number of heavy atoms), and fraction of hydrogen bond acceptors and donors ((HBA + HBD) divided by number of heavy atoms). Figure S1 shows the distribution of the selected ligands along three physicochemical axes used for stratification analysis (Appendix J).

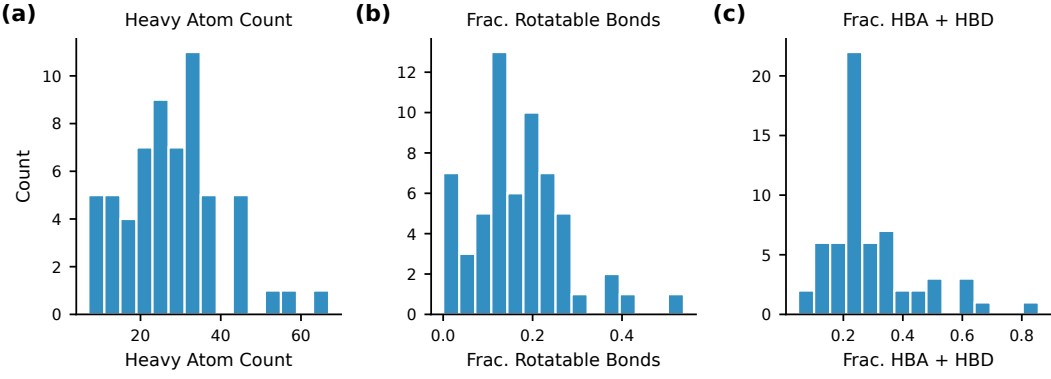

Figure S1: **Distribution of ligand properties in nativeval** ($n = 61$). **(a)** Heavy atom count. **(b)** Fraction of rotatable bonds. **(c)** Fraction of hydrogen bond acceptors and donors (HBA+HBD). Dashed lines indicate tercile boundaries used for ligand property stratification (Appendix J).

**Denovoval curation.**    Because de novo backbones introduce a distribution shift relative to native structures, we used the same CCD codes selected for nativeval and generated de novo backbones with RFD3 (Butcher et al., 2025). For each CCD code, we generated 5 backbone samples at each of 4 chain lengths (150, 250, 350, and 450 residues), yielding $61 \times 4 \times 5 = 1{,}220$ de novo protein-ligand complexes.

## B  DESIGNABLE SUBSET CURATION

The designable subsets are used exclusively for pocket model evaluation and pocket distance selection, where the scaffold sequence has good foldability so that we can evaluate the quality of the pocket sequence in isolation.

**Nativeval-designable** ($n = 20$).  We sampled four sequences per complex using the scaffold model and selected the sequence with the highest $C\alpha$ pLDDT for evaluation. Complexes where the best scaffold-designed sequence achieves strong AF3 self-consistency were retained, using length-dependent thresholds to account for the increased difficulty of structure prediction on longer proteins: scRMSD $\leq 2\,\text{Å}$ and $C\alpha$ pLDDT $\geq 80$ for chains with length $\leq 250$, or scRMSD $\leq 3\,\text{Å}$ and $C\alpha$ pLDDT $\geq 70$ for longer chains.

**Denovoval-designable** ($n = 727$).  We sampled two sequences per complex using the scaffold model and selected the sequence with the highest $C\alpha$ pLDDT. At lengths 150 and 250, each (CCD code, length) combination produced three designable samples. At lengths 350 and 450, some combinations yielded only one or two designable samples due to the increased difficulty of designing longer proteins. We included all available designable samples, resulting in 727 total complexes.

## C  TRAINING DETAILS

**Model configurations.**  The scaffold model (NC$\alpha$COC$\beta$, $\sigma = 0.5\,\text{Å}$) follows the Caliby architecture with $k = 48$ protein residue neighbors and adds ligand conditioning with $m = 16$ nearest ligand atoms and sidechain atoms, with a maximum of 512 tokens and 4,608 atoms per sample. The pocket model (NC$\alpha$COC$\beta$, $\sigma = 0.1\,\text{Å}$) uses $k = 24$ protein residue neighbors, $m = 32$ nearest ligand atoms, and $n = 16$ nearest sidechain atoms, with a maximum of 768 tokens and 4,608 atoms per sample.

**Optimization.**  Both models are trained with Adam ($\beta_1 = 0.9$, $\beta_2 = 0.98$, $\epsilon = 10^{-9}$) using the Noam learning rate schedule with a warmup of 4,000 steps and a scaling factor of 2, yielding a peak learning rate of approximately $2.8 \times 10^{-3}$ at step 4,000. We use a per-GPU batch size of 8 with 4 gradient accumulation steps (effective batch size 32), bf16 mixed precision, and EMA with a decay of 0.99 (Shuai et al., 2025). All models are trained on a single NVIDIA H100 80GB GPU, reaching approximately 80,000 steps in one day.

**Model selection.**  The scaffold model is selected at step 22,500, as foldability degrades with further training. The pocket model is selected at step 82,500, as ligand metrics continue to improve with longer training. This opposite trend in optimal training duration reflects the trade-off between foldability and ligand binding quality discussed in Section 5.1.

## D  EVALUATION DETAILS

**Evaluation budget.**  Because we report best-of-$N$ medians, more evaluations increase the chance of finding a better design, making this effectively a search problem. To ensure fair comparison, we match the number of AF3 evaluations across all methods within each benchmark. For baseline models (ProteinMPNN, LigandMPNN) and the L-Caliby scaffold and pocket models, we sample four sequences per complex on nativeval (and nativeval-designable) and two sequences per complex on denovoval (and denovoval-designable). For two-stage models (L-Caliby Pocket→Scaffold and L-Caliby Scaffold→Pocket), we sample two pocket sequences and two scaffold sequences on nativeval ($2 \times 2 = 4$ total), and two pocket sequences and one scaffold sequence on denovoval ($2 \times 1 = 2$ total).

**AF3 evaluation.**  We used AlphaFold 3 v3.0.1 (Abramson et al., 2024) (commit `a8ecdb2d`). All evaluations were run in single-sequence mode (no multiple sequence alignment, no backbone template) with 10 recycles and 5 diffusion samples, selecting the best model by $C\alpha$ pLDDT. The default model checkpoint was used throughout.

**PoseBusters.** Chemical validity of predicted ligand poses was assessed using PoseBusters v0.6.0 (Buttenschoen et al., 2024).

**Reporting conventions.** All metrics except sequence recovery are reported as medians of the best-of-$N$ samples, and sequence recovery is reported as the median over all samples.

## E SCAFFOLD MODEL SELECTION

The scaffold model was selected from the pool of ligand-conditioned Potts models by evaluating configurations across backbone representations and noise levels on both nativeval and denovoval. The selection criterion prioritized models that achieve strong foldability on denovoval while maintaining adequate performance on nativeval, as the scaffold model must generalize across both native and de novo backbones. Figure S2 shows foldability on denovoval under varying backbone representations: unlike on nativeval (Figure 2), the C$\alpha$-only model achieves the worst foldability, while richer representations converge to superior performance. The NC$\alpha$COC$\beta$ model with $\sigma = 0.5\,\text{Å}$ was selected, as it achieves the highest foldability on denovoval while performing adequately on nativeval.

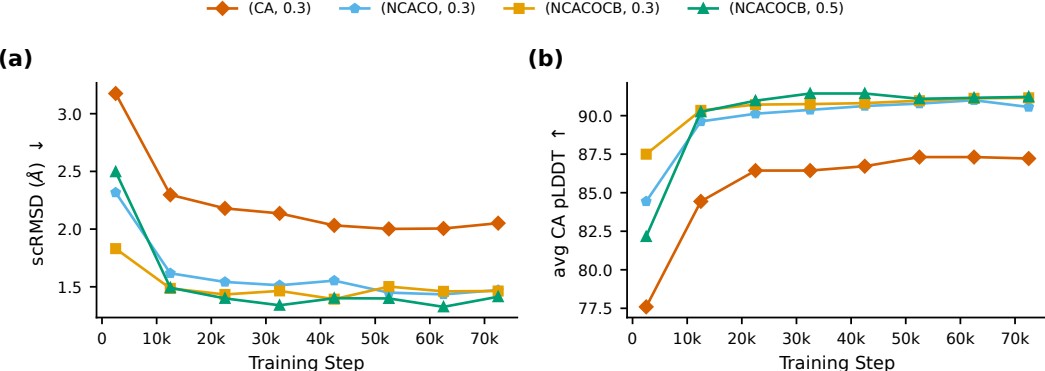

Figure S2: **Foldability of ligand-conditioned Potts models on denovoval.** Backbone representation comparison: unlike on nativeval (Figure 2), C$\alpha$ (blue) achieves the worst foldability, while richer representations (NC$\alpha$CO, NC$\alpha$COC$\beta$) converge to substantially better scRMSD and C$\alpha$ pLDDT.

**Comparison with Caliby.** We compared the selected scaffold model against Caliby (Shuai et al., 2025) on denovoval: the scaffold model achieved comparable performance (scRMSD 1.41 Å vs. 1.20 Å, C$\alpha$ pLDDT 91.0 vs. 89.7), and was adopted as the scaffold component. On nativeval, direct comparison with Caliby was omitted because a subset of nativeval complexes overlap with Caliby's training data, making the comparison unfair.

## F POCKET DISTANCE SELECTION

We swept the pocket distance cutoff $d$ on designable subsets to select the optimal value for the two-stage paradigm. The sweep was performed using a single ligand-conditioned Potts model (NC$\alpha$COC$\beta$, $\sigma = 0.3\,\text{Å}$) for the pocket stage combined with the scaffold model, rather than the final two-stage configuration. On nativeval-designable ($n = 20$), we evaluated $d \in \{4, 5, 6\}\,\text{Å}$ using all-atom distances. On denovoval-designable ($n = 727$), we evaluated $d \in \{6, 7, 8\}\,\text{Å}$ using pseudo-C$\beta$ coordinates, as RFD3-generated backbones lack reliable sidechain geometry.

Table S1: **Pocket distance sweep on nativeval-designable** ($n = 20$). Ligand metrics use pocket-aligned ligand RMSD $\leq 2$ Å and ligand pLDDT $\geq 80$ cutoffs.

| Metric | $d = 4$ Å | $d = 5$ Å | $d = 6$ Å |
|---|---|---|---|
| Placement rate (%) | 20.0 | 25.0 | **35.0** |
| Success rate (%) | 15.0 | 20.0 | **20.0** |

Table S2: **Pocket distance sweep on denovoval-designable** ($n = 727$). Pocket residues are defined using pseudo-C$\beta$ coordinates. Ligand metrics use pocket-aligned ligand RMSD $\leq 2$ Å and ligand pLDDT $\geq 80$ cutoffs.

| Metric | $d = 6$ Å | $d = 7$ Å | $d = 8$ Å |
|---|---|---|---|
| Placement rate (%) | 5.8 | 7.6 | **8.9** |
| Success rate (%) | 3.3 | 4.5 | **6.2** |

## G    SENSITIVITY TO LIGAND EVALUATION CUTOFFS

Tables S3 and S4 report ligand placement metrics at ligand RMSD cutoffs of 2 and 3 Å combined with ligand pLDDT cutoffs of 70 and 80 on nativeval and denovoval, respectively.

On nativeval (Table S3), the rankings are stable across all cutoff combinations, as all successful placements have ligand RMSD $\leq 2$ Å and ligand pLDDT $\geq 90$. The primary cutoff (RMSD $\leq 2$ Å, pLDDT $\geq 80$) used in the main text and the loosest cutoff (RMSD $\leq 3$ Å, pLDDT $\geq 70$) yield identical rankings. L-Caliby (P$\rightarrow$S) achieves the highest placement and success rates under both settings.

Table S3: **Ligand placement metrics on nativeval** ($n = 61$) **at varying RMSD and pLDDT cutoffs.** Placement rate and success rate are reported as fractions. PB Succ: success rate after PoseBusters filtering (available for L-Caliby (P$\rightarrow$S) and LigandMPNN only). Cond. pLDDT reports the median ligand pLDDT among designs meeting the criterion. Best values per cutoff are **bolded**.

| Cutoff | Method | Place Rate | Succ Rate | PB Succ Rate | Cond. pLDDT (Place) |
|---|---|---|---|---|---|
| | Native | 1/61 | 1/61 | – | 96.9 |
| | LigandMPNN | 3/61 | 2/61 | 1/61 | 91.2 |
| RMSD $\leq 2$ Å | L-Caliby (Scaffold) | 4/61 | 4/61 | – | 93.8 |
| pLDDT $\geq 80$ | L-Caliby (Pocket) | 6/61 | 5/61 | – | 89.1 |
| | L-Caliby (Nat. Pkt) | 7/61 | 6/61 | – | 90.1 |
| | L-Caliby (P$\rightarrow$S) | **10/61** | **9/61** | **9/61** | 89.2 |
| | Native | 1/61 | 1/61 | – | 96.9 |
| | LigandMPNN | 3/61 | 3/61 | 1/61 | 91.2 |
| RMSD $\leq 3$ Å | L-Caliby (Scaffold) | 4/61 | 4/61 | – | 93.8 |
| pLDDT $\geq 70$ | L-Caliby (Pocket) | 6/61 | 5/61 | – | 89.1 |
| | L-Caliby (Nat. Pkt) | 7/61 | 6/61 | – | 90.1 |
| | L-Caliby (P$\rightarrow$S) | **10/61** | **9/61** | **9/61** | 89.2 |

On denovoval, the best-performing model varies with the cutoff. At the primary cutoff (RMSD $\leq 2$ Å, pLDDT $\geq 80$), the pocket model achieves the highest placement rate (5.5%), success rate (3.9%), and PB-valid success rate (3.6%). At the looser cutoff (RMSD $\leq 3$ Å, pLDDT $\geq 70$), P$\rightarrow$S achieves the highest placement rate (10.2%) and all three L-Caliby and LigandMPNN models achieve comparable success rates (8.0%). CCD coverage shows that the pocket model covers the most unique ligands across all cutoff combinations.

Table S4: **Ligand placement metrics on denovoval** ($n = 1{,}220$) **at varying RMSD and pLDDT cutoffs.** Succ: success rate (%). PB Succ: PB-valid success rate (%). CCD Cov: fraction of 61 CCD codes with at least one success. CCD Cov (PB): CCD coverage after PoseBusters filtering. PB metrics are available for L-Caliby (P→S), L-Caliby (Pocket), and LigandMPNN. Best values per cutoff are **bolded**.

| Cutoff | Method | Place (%) | Succ (%) | PB Succ (%) | CCD Cov | CCD Cov (PB) | Cond. pLDDT (Succ) |
|---|---|---|---|---|---|---|---|
| RMSD $\leq 2$ Å pLDDT $\geq 70$ | RFD3 | 0.2 | 0.2 | – | – | – | 82.7 |
| | LigandMPNN | 4.9 | 4.5 | 3.5 | 35/61 | 29/61 | 84.4 |
| | L-Caliby (Scaffold) | 1.1 | 0.7 | – | – | – | 77.3 |
| | L-Caliby (Pocket) | **5.5** | **5.1** | **4.6** | **37/61** | **33/61** | 87.8 |
| | L-Caliby (P→S) | 5.3 | 4.7 | 4.2 | 31/61 | 30/61 | 84.6 |
| RMSD $\leq 2$ Å pLDDT $\geq 80$ | RFD3 | 0.2 | 0.1 | – | – | – | 89.6 |
| | LigandMPNN | 4.9 | 3.1 | 2.5 | 27/61 | 21/61 | 89.3 |
| | L-Caliby (Scaffold) | 1.1 | 0.2 | – | – | – | 91.1 |
| | L-Caliby (Pocket) | **5.5** | **3.9** | **3.6** | **32/61** | **30/61** | 90.3 |
| | L-Caliby (P→S) | 5.3 | 3.4 | 3.2 | 26/61 | 26/61 | 88.2 |
| RMSD $\leq 3$ Å pLDDT $\geq 70$ | RFD3 | 0.9 | 0.7 | – | – | – | 77.3 |
| | LigandMPNN | 9.1 | **8.0** | 6.3 | 45/61 | 39/61 | 84.0 |
| | L-Caliby (Scaffold) | 5.6 | 3.9 | – | – | – | 77.8 |
| | L-Caliby (Pocket) | 9.6 | **8.0** | **7.2** | **47/61** | **43/61** | 84.4 |
| | L-Caliby (P→S) | **10.2** | **8.0** | **7.2** | 45/61 | 42/61 | 83.7 |
| RMSD $\leq 3$ Å pLDDT $\geq 80$ | RFD3 | 0.9 | 0.2 | – | – | – | 89.6 |
| | LigandMPNN | 9.1 | **5.5** | 4.4 | **38/61** | 30/61 | 87.7 |
| | L-Caliby (Scaffold) | 5.6 | 1.4 | – | – | – | 87.0 |
| | L-Caliby (Pocket) | 9.6 | 5.1 | 4.7 | **38/61** | **36/61** | 89.4 |
| | L-Caliby (P→S) | **10.2** | 5.2 | **4.8** | 34/61 | 33/61 | 86.2 |

# H LIGAND-CONDITIONING EFFECT AND METRIC SELECTION

To isolate the effect of ligand conditioning, we compared a ligand-conditioned model against a protein-only model with identical architecture, backbone representation, noise level, and training step (NC$\alpha$COC$\beta$, $\sigma = 0.5$ Å, step 22500) on nativeval. Table S5 reports the comparison at the ligand RMSD $\leq 5$ Å and ligand pLDDT $\geq 70$ cutoff.

When comparing metrics, the protein-only model achieves a lower median ligand RMSD (13.5 Å vs. 14.9 Å) and higher median ligand pLDDT (52.8 vs. 49.1). However, the ligand-conditioned model achieves substantially higher placement rate (21.3% vs. 9.8%) and ligand success rate (16.4% vs. 6.6%), as well as higher sequence recovery at all pocket distance cutoffs. Furthermore, Table S3 shows that even the native sequence achieves only 1 successful placement, yet with very high conditional pLDDT. This illustrates that the median ligand RMSD, ligand pLDDT, and conditional pLDDT can be misleading when the number of successful placements is small. We therefore adopt ligand placement rate and ligand success rate as the primary ligand metrics throughout this work.

Table S5: **Ligand-conditioned vs. protein-only model on nativeval** ($n = 61$). LC = ligand-conditioned (NC$\alpha$COC$\beta$, $\sigma = 0.5$ Å, step 22500), PO = protein-only (NC$\alpha$COC$\beta$, $\sigma = 0.5$ Å, step 22500). Ligand metrics use RMSD $\leq 5$ Å and pLDDT $\geq 70$. Seq. Rec. = sequence recovery. Pkt = pocket.

| | scRMSD (Å) | C$\alpha$ pLDDT | Lig RMSD | Lig pLDDT | Place Rate | Succ Rate | Seq Rec | Pkt 4Å Rec | Pkt 5Å Rec | Pkt 6Å Rec |
|---|---|---|---|---|---|---|---|---|---|---|
| LC | 5.52 | 77.4 | 14.9 | 49.1 | **21.3%** | **16.4%** | 41.6% | 45.3% | 44.3% | 44.6% |
| PO | **5.89** | **81.1** | **13.5** | **52.8** | 9.8% | 6.6% | 39.8% | 36.4% | 35.7% | 37.9% |

# I POSEBUSTERS VALIDATION

We evaluated all AF3-predicted ligand poses using PoseBusters (Buttenschoen et al., 2024), which checks chemical validity criteria including bond lengths, bond angles, internal steric clashes, and

molecular strain, independent of the structure prediction model. We apply PoseBusters analysis to L-Caliby (P→S), L-Caliby (Pocket), and LigandMPNN.

Table S6 reports PoseBusters pass rates across datasets and cutoff combinations. On nativeval, L-Caliby (P→S) achieves 100% PB validity across all cutoffs, though the small number of successful designs (9 at the primary cutoff) limits statistical power. On denovoval, L-Caliby (P→S) consistently achieves higher PB validity than LigandMPNN: at the primary cutoff (RMSD $\leq 2$ Å, pLDDT $\geq 80$), L-Caliby (P→S) achieves 92.9% PB validity (39/42) compared to LigandMPNN's 78.9% (30/38). This pattern holds across all cutoff combinations, with L-Caliby (P→S) PB validity ranging from 87.1% to 92.9% while LigandMPNN ranges from 77.5% to 80.6%.

Notably, PB validity increases with stricter ligand pLDDT thresholds for both methods on denovoval: at RMSD $\leq 2$ Å, L-Caliby (P→S) PB validity rises from 87.7% (placement) to 89.5% (pLDDT $\geq 70$) to 92.9% (pLDDT $\geq 80$). This suggests that AF3 ligand confidence is partially informative of chemical validity, though the correlation is imperfect given the gap between AF3 and PB assessments.

At the CCD code level, PB filtering has a disproportionate effect on LigandMPNN. At the primary cutoff, L-Caliby (P→S) covers 42.6% of CCD codes (26/61) before PB filtering and retains all 26 after filtering (42.6%), while LigandMPNN covers 44.3% (27/61) before but drops to 34.4% (21/61) after PB filtering. This indicates that LigandMPNN's successful ligand placements are more frequently chemically invalid, and that its apparent CCD coverage advantage over L-Caliby (P→S) reverses after PB validation.

Table S6: **PoseBusters validation of L-Caliby (P→S) and LigandMPNN across datasets and cutoffs.** $n_{\text{sel}}$: number of designs meeting the AF3 cutoff. PB Valid: fraction passing all PoseBusters chemical validity checks. Lig Succ & PB Valid: fraction of all complexes that meet both the AF3 ligand success criterion and PoseBusters chemical validity.

| Dataset | Cutoff | L-Caliby (P→S) | | LigandMPNN | |
| --- | --- | --- | --- | --- | --- |
| | | $n_{\text{sel}}$ / PB Valid | Lig Succ & PB Valid | $n_{\text{sel}}$ / PB Valid | Lig Succ & PB Valid |
| Nativeval ($n = 61$) | RMSD $\leq 2$ (placement) | 10 / 100.0% | 16.4% | 3 / 33.3% | 1.6% |
| | RMSD $\leq 2$, pLDDT $\geq 80$ | 9 / 100.0% | 14.8% | 2 / 50.0% | 1.6% |
| | RMSD $\leq 3$, pLDDT $\geq 70$ | 9 / 100.0% | 14.8% | 3 / 33.3% | 1.6% |
| Denovoval ($n = 1,220$) | RMSD $\leq 2$ (placement) | 65 / 87.7% | 4.7% | 60 / 78.3% | 3.9% |
| | RMSD $\leq 2$, pLDDT $\geq 70$ | 57 / 89.5% | 4.2% | 55 / 78.2% | 3.5% |
| | RMSD $\leq 2$, pLDDT $\geq 80$ | 42 / 92.9% | 3.2% | 38 / 78.9% | 2.5% |
| | RMSD $\leq 3$ (placement) | 124 / 87.1% | 8.9% | 111 / 77.5% | 7.0% |
| | RMSD $\leq 3$, pLDDT $\geq 70$ | 98 / 89.8% | 7.2% | 97 / 79.4% | 6.3% |
| | RMSD $\leq 3$, pLDDT $\geq 80$ | 63 / 92.1% | 4.8% | 67 / 80.6% | 4.4% |

## J    BREAKDOWN ANALYSIS

We examine how design success varies by ligand properties (Table S7) and protein length (Table S8) on denovoval, comparing L-Caliby (P→S), L-Caliby (Pocket), and LigandMPNN. Success is measured at the CCD code level: for each ligand, we ask whether at least one of the associated de novo complexes achieves successful ligand placement. We report results at two cutoff combinations to capture both the primary setting (RMSD $\leq 2$ Å, pLDDT $\geq 80$) and a more lenient setting (RMSD $\leq 3$ Å, pLDDT $\geq 70$) that provides higher statistical power.

**Ligand property stratification.**    We stratify the 61 CCD codes into terciles along three physico-chemical axes: heavy atom count (molecular size), fraction of rotatable bonds (number of rotatable bonds divided by number of heavy atoms, measuring flexibility), and fraction of hydrogen bond acceptors and donors ((HBA + HBD) divided by number of heavy atoms, measuring polarity). Table S7 reports CCD-level success rates within each bin.

At the primary cutoff, the pocket model consistently achieves the highest or comparable success rates across most bins, outperforming both P→S and LigandMPNN. This is especially notable for medium-sized ligands (22–32 heavy atoms: Pocket 73.7%, P→S 52.6%, LMPNN 52.6%)

and medium-flexibility ligands (0.125–0.205 rotatable bond fraction: Pocket 68.4%, P→S 52.6%, LMPNN 47.4%). At the lenient cutoff, L-Caliby (P→S) tends to outperform LigandMPNN on medium-sized ligands (94.7% vs. 68.4%) and low-polarity ligands (85.7% vs. 76.2%), while LigandMPNN performs better on small ligands (90.5% vs. 81.0%). All methods show reduced success on large ligands (>32 heavy atoms) and polar ligands (high HBA+HBD fraction), consistent with the increased difficulty of placing larger and more polar molecules.

Table S7: **CCD-level success rate by ligand property on denovoval.** The 61 CCD codes are split into terciles along each axis. Success rate: fraction of CCD codes with at least one successful design at the given cutoff.

| Axis | Bin | RMSD $\leq$ 2, pLDDT $\geq$ 80 | | | RMSD $\leq$ 3, pLDDT $\geq$ 70 | | |
|---|---|---|---|---|---|---|---|
| | | P→S | Pocket | LMPNN | P→S | Pocket | LMPNN |
| Heavy atom count | 7–22 ($n = 21$) | 52.4% | 57.1% | 33.3% | 81.0% | 90.5% | 90.5% |
| | 22–32 ($n = 19$) | 52.6% | 73.7% | 52.6% | 94.7% | 89.5% | 68.4% |
| | 32–67 ($n = 21$) | 23.8% | 33.3% | 47.6% | 47.6% | 52.4% | 61.9% |
| Frac rot. bonds | 0–0.125 ($n = 21$) | 38.1% | 47.6% | 47.6% | 71.4% | 76.2% | 76.2% |
| | 0.125–0.205 ($n = 19$) | 52.6% | 68.4% | 47.4% | 73.7% | 78.9% | 63.2% |
| | 0.205–0.538 ($n = 21$) | 38.1% | 47.6% | 38.1% | 76.2% | 76.2% | 81.0% |
| Frac HBA+HBD | 0.047–0.231 ($n = 21$) | 57.1% | 66.7% | 52.4% | 85.7% | 76.2% | 76.2% |
| | 0.231–0.310 ($n = 19$) | 42.1% | 52.6% | 63.2% | 68.4% | 89.5% | 84.2% |
| | 0.310–0.857 ($n = 21$) | 28.6% | 42.9% | 19.0% | 66.7% | 66.7% | 61.9% |

**Protein length breakdown.** Table S8 reports success rates stratified by protein length. All methods show a strong length dependence, with performance dropping sharply at longer lengths. At 150 residues and the primary cutoff, the pocket model achieves the highest success rate (45.9%), outperforming both P→S (36.1%) and LigandMPNN (37.7%). This supports the observation from Section 5.4 that the pocket model's overall advantage on denovoval is primarily driven by short proteins, where the reduced fold complexity allows the pocket model alone to produce foldable sequences. At 250 residues, P→S and the pocket model achieve comparable success rates (both 14.8%), while at 350 and 450 residues, success at the primary cutoff is near zero for all methods. We additionally report PB-valid success rates, which show that PoseBusters filtering has minimal additional impact at shorter lengths but removes a larger fraction of LigandMPNN successes at longer lengths: at 250 residues and the primary cutoff, LigandMPNN drops from 11.5% to 6.6% while P→S retains 14.8%.

Table S8: **Success rate by protein length on denovoval (per CCD code, $n = 61$ per length).** Placement: ligand RMSD $\leq$ cutoff. Succ: placement with ligand pLDDT $\geq$ cutoff. PB: success with PoseBusters validity.

| Cutoff | Length | L-Caliby (P→S) | | | L-Caliby (Pocket) | | | LigandMPNN | | |
|---|---|---|---|---|---|---|---|---|---|---|
| | | Place | Succ | PB | Place | Succ | PB | Place | Succ | PB |
| RMSD $\leq$ 2 pLDDT $\geq$ 80 | 150 | 41.0% | 36.1% | 36.1% | 55.7% | 45.9% | 39.3% | 49.2% | 37.7% | 31.1% |
| | 250 | 27.9% | 14.8% | 14.8% | 21.3% | 14.8% | 13.1% | 18.0% | 11.5% | 6.6% |
| | 350 | 4.9% | 0.0% | 0.0% | 1.6% | 0.0% | 0.0% | 8.2% | 3.3% | 3.3% |
| | 450 | 3.3% | 0.0% | 0.0% | 4.9% | 3.3% | 3.3% | 4.9% | 1.6% | 0.0% |
| RMSD $\leq$ 3 pLDDT $\geq$ 70 | 150 | 68.9% | 65.6% | 62.3% | 65.6% | 65.6% | 54.1% | 70.5% | 68.9% | 57.4% |
| | 250 | 41.0% | 29.5% | 29.5% | 32.8% | 32.8% | 27.9% | 36.1% | 29.5% | 23.0% |
| | 350 | 19.7% | 11.5% | 11.5% | 21.3% | 18.0% | 13.1% | 19.7% | 14.8% | 11.5% |
| | 450 | 9.8% | 4.9% | 3.3% | 4.9% | 4.9% | 4.9% | 8.2% | 4.9% | 1.6% |

