# OpenReview forum: "Two-Stage Ligand-Protein Complex Sequence Design with L-Caliby"
_ICLR.cc/2026/Workshop/FM4Science — ICLR 2026 Workshop FM4Science Poster_

### Official Review · Reviewer_E3Yp · 2026-02-21
**Review on Ligand-Conditioned Protein Sequence Design with L-Caliby**

**Rating:** 7
**Confidence:** 4

**Review:**

# Quality

The technical approach is sound and well-motivated. The paper correctly identifies a gap that Caliby's Potts model achieves superior self-consistency for protein-only design, but cannot handle ligands while LigandMPNN handles ligands but suffers from poor self-consistency. The integration is natural and well-executed.

# Clarity

The paper is well-written and clearly organized:

-  The architecture description (Section 2.1) is concise, and Figure 1 provides a helpful visual overview. The pipeline is easy to follow.

- The metrics section (Section 2.4) is thorough and well-defined, with clear explanations of each evaluation axis. The distinction between pocket-aligned ligand RMSD and standard ligand RMSD is important and well-explained.

- The training curves in Figure 3 effectively communicate the Cα-only advantage, showing clear overfitting patterns for NCαCO and NCαCOCβ variants.

# Originality

The core novelty is the integration of Caliby's Potts model and LigandMPNN's ligand encoder into a unified framework.

# Significance

This work addresses a real and important bottleneck in computational protein design which is the disconnect between high native sequence recovery and poor self-consistency in ligand-conditioned sequence design. The practical significance is high for the protein design.

# Pros

- Strong empirical improvements on the key metric of ligand placement success rate. The 8× improvement in ligand placement and improvement in scRMSD are substantial.

- The Cα-only training finding is well-supported by the training curves (Figure 3).

- Using AF3 in single-sequence mode is especially well-motivated.

# Cons

- The evaluation relies entirely on AF3 self-consistency as a proxy for designability.

- The absolute ligand placement success rates remain low.

- The paper does not compare against other recent structure-conditioned sequence design methods.

---

### Official Review · Reviewer_egm4 · 2026-02-25
**Review of "Ligand-Conditioned Protein Sequence Design with L-Caliby"**

**Rating:** 7
**Confidence:** 3

**Review:**

This paper introduces **L-Caliby**, a ligand-conditioned Potts-model-based protein sequence design method that aims to improve **backbone foldability** and **ligand placement** relative to existing ligand-conditioned autoregressive approaches (notably LigandMPNN). The key idea is to combine the **Potts/Markov random field inductive bias** from Caliby (which has strong AF2 self-consistency in protein-only design) with LigandMPNN’s **protein–ligand interaction encoder**, and evaluate designs using **AlphaFold3 (AF3) single-sequence self-consistency**, including ligand pose prediction. On held-out native ligand–protein complexes and RFdiffusion3-generated de novo complexes with unseen ligands, L-Caliby reports substantially higher ligand placement success rates and lower scRMSD than LigandMPNN.

### Summary of method and claims
**Architecture.** L-Caliby combines (i) a **Potts / Markov random field** sequence model head (as in Caliby) with (ii) a **protein–ligand interaction encoder** (as in LigandMPNN). The model outputs single-site fields $h_i$ and pairwise couplings $J_{ij}$ and defines:

$$
P(s \\mid X, L) \\propto \\exp\\Big( \\sum_i h_i(s_i) + \\sum_{i<j} J_{ij}(s_i, s_j) \\Big)
$$

where $s$ is the amino-acid sequence, $X$ the backbone, and $L$ ligand coordinates/features. Sequences are sampled via **discrete Langevin Monte Carlo (DLMC)**.

**What the paper emphasizes.** Two training design choices are highlighted as important for generalization: **C$\\alpha$-only** backbone inputs (plus coordinate noise) and **cluster-balanced sampling** across protein families / interfaces.

**Evaluation.** Designs are evaluated with **AF3 single-sequence self-consistency** (no MSA/templates) on (i) 147 held-out native complexes and (ii) 1,000 RFdiffusion3 de novo complexes, using scRMSD and pocket-aligned ligand RMSD (success if RMSD $\\le 5$ \\AA), plus pocket sequence recovery.

**Main empirical takeaway.** Relative to LigandMPNN, L-Caliby reports substantially better AF3 self-consistency and higher ligand-placement success (notably 11.2% vs 1.4% on native complexes; 12.8% vs 10.4% on de novo complexes).

### Strengths
- **Well-motivated architectural synthesis**: combining the Potts-model inductive bias (pairwise couplings) with explicit ligand conditioning is a natural next step given Caliby’s strong protein-only self-consistency and LigandMPNN’s ligand awareness.
- **Evaluation matches the deployment goal**: using AF3 to jointly assess foldability and ligand pose is appropriate for ligand-binding design, and single-sequence mode reduces reliance on evolutionary retrieval signals.
- **Clear and relevant metrics**: separating scRMSD, ligand RMSD, and pocket recovery helps interpret whether gains come from better folding, better pocket specification, or both.
- **Concrete, reproducible engineering insights**: C$\\alpha$-only training and cluster-balanced sampling are actionable contributions that could transfer to other models.

### Weaknesses and concerns (major)
- **Absolute ligand-placement success remains low (and the threshold is lenient)**: success rates around 10–13% at a 5 \\AA pocket-aligned RMSD threshold indicate the problem is far from solved. Given the importance of pose accuracy, the paper should report success at stricter cutoffs (2/3/4/5 \\AA) and provide full distributions (not just medians).
- **AF3 is the single evaluation oracle**: AF3 single-sequence mode is a reasonable attempt to reduce evolutionary retrieval effects, but it remains a learned predictor with unknown biases and calibration. Without orthogonal validation (docking/relaxation, steric clashes/strain checks, binding-site chemistry plausibility), it is difficult to conclude the improvements correspond to real binding feasibility rather than AF3-preference alignment.
- **Exclusions / failure handling can bias comparisons**: the paper notes that structural alignment failed for some LigandMPNN/ProteinMPNN designs on native complexes and excludes those from metrics. These should arguably be counted as failures (worst-case scRMSD / zero success), or at least results should be reported both ways.
- **One sampled sequence per complex is not a stable design protocol**: Potts/DLMC sampling can have high variance. Reporting only one draw per target may understate achievable performance (best-of-$k$) or overstate it (if cherry-picked checkpoints). The paper should report variance across multiple sampled sequences per target and best-of-$k$ trade-offs.
- **Attribution is still somewhat entangled**: L-Caliby improves scRMSD substantially vs LigandMPNN, and better folding can mechanically improve ligand placement. The paper would be stronger with analyses that condition on foldability (e.g., ligand RMSD/success among designs with scRMSD below a threshold), to isolate the incremental value of ligand conditioning in the Potts framework.
- **Compute/inference cost is under-discussed**: DLMC sampling can be materially slower than autoregressive decoding. Practical adoption needs sampling-step counts, time per design, and sensitivity to DLMC hyperparameters (and how many sequences are typically needed per target).

### Weaknesses (minor)
- **C$\\alpha$-only training claim is strong**: the paper makes a compelling empirical point, but the mechanism remains speculative. More controlled tests (e.g., removing pairwise couplings, or matching capacity/training schedules across backbone-input variants) would strengthen the interpretation.
- **Ligand pLDDT ambiguity**: the paper itself notes that ligand pLDDT may be a noisy metric; relying on it for nuanced comparisons should be done cautiously.

### Suggestions for improvement
- **Report stricter ligand-pose metrics**: success at 2/3/4/5 \\AA, plus distributions (CDF/quantiles) for both ligand RMSD and scRMSD.
- **Add orthogonal validation signals**: docking/relaxation or simple plausibility checks (clashes, strain, pocket volume/shape compatibility) on a representative subset.
- **Report failure-inclusive metrics**: include non-alignable AF3 predictions as failures (or provide dual reporting: excluded vs counted-as-fail) for fair baseline comparison.
- **Sample multiple sequences per target**: report mean/median and best-of-$k$ performance as $k$ increases (and the compute cost) to reflect realistic design usage.
- **Improve attribution with targeted ablations**:
  - compare L-Caliby to a Potts model trained on interfaces but with ligand inputs ablated at inference,
  - test whether removing pairwise couplings disproportionately hurts C$\\alpha$-only performance (as proposed in the discussion),
  - analyze cases where scRMSD is good but ligand placement fails vs succeeds (and vice versa).
- **Include runtime/cost reporting**: DLMC sampling steps, time per design, and comparison to LigandMPNN inference time.
- **Benchmark diversity**: stratify results by ligand novelty (Tanimoto bins), ligand size/rotatable bonds, and backbone length (some breakdown exists; expanding it would help).

### Summary - Pros and cons

**Pros**
- **Strong workshop-relevant idea**: brings Potts-model inductive bias to ligand-conditioned design.
- **Appropriate AF3-based evaluation** for both folding and ligand placement, with single-sequence controls.
- **Meaningful improvements over LigandMPNN** on both native and de novo benchmarks.
- **Actionable training insights** (C$\\alpha$-only, cluster-balanced sampling).

**Cons**
- **Success rates are still low in absolute terms**, and additional stricter cutoffs/stratifications are needed to judge practical impact.
- **Evaluation depends heavily on AF3**, with limited orthogonal validation of binding feasibility.
- **Reported comparisons may be biased by failure exclusions** unless non-alignable predictions are counted as failures (or dual-reported).
- **Single-sample evaluation** (one designed sequence per target) does not reflect typical best-of-$k$ design usage and hides variance.
- **Compute cost and sampling details** for DLMC are not fully surfaced.
- **Attribution of gains** (ligand conditioning vs foldability improvements) could be clearer.

---

### Official Review · Reviewer_kNN7 · 2026-02-25
**The motivation is sensible and technically grounded, but the paper would benefit from clearer organization.**

**Rating:** 4
**Confidence:** 2

**Review:**

This paper proposes L-Caliby, a ligand-conditioned Potts model for protein sequence design. The work extends Caliby (a Potts-based protein design model) by integrating ligand-aware representations inspired by LigandMPNN. While the paper is technically coherent and shows empirical improvements, several conceptual, experimental, and presentation issues limit the strength of its conclusions.

Pros:
1. Clear motivation and technically sound architecture. The integration of a Potts-based sequence model with ligand-aware encoding is logically consistent and technically reasonable.
2. The observation that Cα-only backbone training improves generalization and mitigates overfitting is interesting and potentially impactful.

Cons:
1. Difficult to follow due to issues with the logic flow, missing explanation of the results. There are places where the logical flow is confusing. For example, the paper would benefit significantly from a clearer statement of contributions in the Introduction. Also, in Section 2.2 (methodology), the authors reference results from Section 3.2 (e.g., around line 108) to justify modeling decisions. In addition, Table 2 is not sufficiently explained in the text. The reader is left to interpret the comparisons without clear guidance. More importantly, Table 2 shows that Caliby sometimes outperforms the proposed ligand-conditioned model on certain metrics, which is not discussed or explained. Does ligand conditioning introduce trade-offs? Are improvements task-specific? How should readers interpret these contradictions? Similar question in Table 4 where both L-Caliby and Cα-Caliby underperform LigandMPNN. Why do these variants fall behind the baseline?
2. Seems Appendix B is missing.

---

### Official Review · Reviewer_aWRW · 2026-02-26
**Clean extension of Potts models to ligand-conditioned sequence design**

**Rating:** 8
**Confidence:** 3

**Review:**

Paper Summary: L-Caliby augments the Caliby Potts model architecture with LigandMPNN's protein-ligand interaction encoder for ligand-conditioned protein sequence design. The model is trained with Cα-only backbone coordinates as a form of structural regularization, and uses cluster-balanced sampling to handle skewed protein family distributions. Evaluated on 147 native and 1,000 de novo protein-ligand complexes via single-sequence AlphaFold3 self-consistency, L-Caliby substantially outperforms LigandMPNN on both backbone foldability and ligand placement.

Strengths:
1) L-Caliby demonstrates clear improvements over LigandMPNN across both benchmarks. On native complexes, median scRMSD drops from 16.34 to 6.45 and ligand placement success jumps from 1.4% to 11.2%. Importantly, the gains also hold on de novo RFdiffusion3 backbones, which is the harder and more practically relevant setting.

2) The comparison between L-Caliby and Cα-Caliby is well controlled. Both share the same architecture and training procedure, with ligand conditioning as the only variable. Placement success jumps from 3.4% to 11.2% on native complexes, pocket sequence recovery improves at all distance cutoffs (Table 4), and backbone foldability is preserved.

Weaknesses:

1) Ligand placement success rates are still low in absolute terms (11-13%). The paper frames these as improvements over LigandMPNN, which is fair, but it would help to contextualize what success rates are achievable given current structure prediction tools. Is this by virtue of the ceiling imposed by AF3's own accuracy?

2) Dovetailing off of weakness 1, the evaluation relies entirely on AF3 in single-sequence mode as the oracle. The authors motivate this choice well, but AF3 has its own biases and failure modes. It would be useful to discuss how confident we should be that the method rankings reflect genuine sequence quality differences rather than quirks of how AF3 handles different sequence distributions.

Small Question:
The paper proposes a clean experiment to test the Cα-only hypothesis by removing pairwise couplings from the Potts head. Are there any preliminary results from this?

---

### Meta-Review · Area_Chair_vTiw · 2026-02-28

**Recommendation:** Accept (Poster)
**Confidence:** 3

**Metareview:**

This paper tackles the challenge of ligand-conditioned protein sequence design by combining Caliby's Potts model with a LigandMPNN's protein-ligand interaction module. Reviewers found this integration to be technically correct and well executed. Additionally, the insights provided about training with a Cα-only backbone were noted as interesting by reviewers and considered a valuable contribution in its own right. The paper has solid clarity and the empirical evidence validates this contribution, showing strong gains over LigandMPNN.
The primary weakness shared by reviewers stems from the heavy reliance on AlphaFold3 for evaluation.

---

### Decision · Program_Chairs · 2026-03-03

Accept (Poster)